# The Combination of Inflammatory Biomarkers as Prognostic Indicator in Salivary Gland Malignancy

**DOI:** 10.3390/cancers14235934

**Published:** 2022-11-30

**Authors:** Vincenzo Abbate, Simona Barone, Stefania Troise, Claudia Laface, Paola Bonavolontà, Daniela Pacella, Giovanni Salzano, Giorgio Iaconetta, Luigi Califano, Giovanni Dell’Aversana Orabona

**Affiliations:** 1Maxillofacial Surgery Unit, Department of Neurosciences, Reproductive and Odontostomatological Sciences, University of Naples Federico II, Via Sergio Pansini 5, 80131 Naples, Italy; 2Department of Public Health, University of Naples Federico II, Via Sergio Pansini 5, 80131 Naples, Italy; 3ENT Department, Istituto Nazionale Tumori IRCCS Fondazione G. Pascale, Via Mariano Semmola 53, 80131 Napoli, Italy; 4Neurosurgery Unit, Department of Medicine, Surgery and Odontoiatrics, University of Salerno, Via Giovanni Paolo II 132, 84084 Salerno, Italy

**Keywords:** malignant salivary gland tumors, systemic immune-inflammation index, platelet-to-lymphocyte ratio, neutrophil-to-lymphocyte ratio, systemic inflammation response index

## Abstract

**Simple Summary:**

Clinical management of Malignant Salivary Gland Tumors is still a challenge for clinicians. Direct and indirect costs for the diagnosis and treatment of these tumors are significant. For this reason, the need to develop a fast and low-cost prognostic system to stratify patients at higher risk appears to be mandatory. The efficacy of inflammatory biomarkers such as the systemic immune-inflammation index (SII), systemic inflammation response index (SIRI), platelet-to-lymphocyte ratio (PLR), and neutrophil-to-lymphocyte ratio (NLR), as prognostic values that are quickly available and low cost, has been confirmed in various fields of oncology. The aim of this study was to investigate the impact of these biomarkers taken individually and combined, to assess the overall survival (OS) in patients surgically treated for malignant salivary gland tumors. This study determined that the combination of SII + SIRI can independently predict the overall survival of patients after surgery for malignant salivary gland tumors.

**Abstract:**

Background: The aim of this study was to investigate how the systemic inflammation response index (SIRI), systemic immune-inflammation index (SII), neutrophil-to-lymphocyte ratio (NLR), and platelet-to-lymphocyte ratio (PLR), taken individually and combined, are associated with overall survival (OS) in patients surgically treated for malignant salivary gland tumors (MSGTs). Methods: A retrospective analysis of 74 cases following surgery at our department between January 2011 and June 2018 was performed. The Receiver Operating Characteristic (ROC) curve was used to calculate the optimal cutoff values for SII, SIRI, PLR, and NLR. Survival curves of different groups at 1–3–5 years were estimated using the Kaplan–Meier method. Results: The optimal thresholds with the highest sensitivity and specificity were 3.95 for NLR, 187.6 for PLR, 917.585 for SII, and 2.045 for SIRI. The ROC curves revealed that the best combination with AUC = 0.884 was SII + SIRI. The estimated 5-year OS probability in patients with SII+ SIRI scores of 0, 1, and 2 was 96%, 87.5% and 12.5%, respectively (*p* < 0.001). Conclusion: SII+ SIRI can independently predict the OS of patients after MSGT surgery. The prognostic score system based on SII+ SIRI may be good clinical practice as a reference for clinical decision-making.

## 1. Introduction

Malignant Salivary Gland Tumors (MSGTs) are rare diseases accounting for up to 20% of all salivary gland neoplasms, representing approximately 1–7% of head and neck cancers and 0.3% of all human cancers [1]. 

The incidence is reported at 0.5–2 per 100,000, mostly in patients between 40 and 60 years old, with a slightly higher number of cases in men than in women. Although the parotid gland is the most common site for salivary neoplasms, only 25% of parotid tumors are malignant compared to 40%, 50%, and 90% of neoplasms reported as malignant in submandibular gland, minor salivary gland, and sublingual gland tumors, respectively [2,3]. 

The wide variations in the tumor etiology, microscopic histology, growth patterns, and tumor characteristics can make the diagnosis and treatment challenging for clinicians. The World Health Organization (WHO) recognized 20 different malignant salivary gland cancers in 2017 [4]. The most common histologies include mucoepidermoid carcinoma (MEC), acinic cell carcinoma (ACC), adenoid cystic carcinoma (AdCC), carcinoma ex-pleomorphic adenoma (CExPA), and adenocarcinoma [5]. 

The reported 5-year survival rate of these tumors is approximately 25% to 80% based on different features such as histology, grading, tumor size, AJCC (American Joint Committee on Cancer) stage [6], extraparenchymal extension, and lymph node involvement, whereas the morbidity rate after surgery is reported to be 30–65%, based on the need to intra-operatively sacrifice the facial nerve [7,8]. The challenge in recent years has been to find quantifiable pre-surgery factors that can predict survival. The importance of specific blood inflammatory biomarkers, such as the neutrophil-to-lymphocyte ratio (NLR), platelet-to-lymphocyte ratio (PLR), Systemic Immune Inflammation Index (SII), and systemic inflammation response index (SIRI), as prognostic values have already been shown in relation to various malignant tumors such as lung adenocarcinomas, oral cancer, early-stage clear-cell carcinomas, endometrioid and mucinous ovarian carcinomas, hepatocellular carcinomas, and others [9,10,11,12].

However, to the best of our knowledge, there are no significant data on the prognostic efficacy of these biomarkers in relation to salivary gland malignant tumors. Thus, the aim of our study has been to investigate the predictive value of these inflammatory biomarkers, considered individually and in combination, in terms of assessing overall survival (OS) in patients surgically treated for malignant salivary gland tumors. 

## 2. Materials and Methods

### 2.1. Study Population

This is a retrospective chart review performed on patients surgically treated for an MSGT in the Maxillofacial Surgery Unit of the University Federico II of Naples between January 2011 and June 2018. We enrolled the patients according to these inclusion criteria: -Previous salivary gland surgery.-Post-operative histopathology-confirmed salivary gland malignant tumor.-No active infection, chronic inflammation, autoimmune disease, or other malignancy at the time of admission.-A mean follow-up period of 5 years.

Patients who met the following criteria were excluded from this study:-Serious complications or death occurring within 15 days of the surgery.

Administration of neoadjuvant chemotherapy or a lack of pre-operative inflammation index data. 

A total of 74 patients were enrolled in this study in accordance with the above inclusion and exclusion criteria. 

### 2.2. Data Collection

The data were collected through patients’ medical records. The analysis was based on demographic information (age, sex), the histological type of the tumor, the site of the tumor, the surgical treatment, and the pathological stage according to the Union for International Cancer Control/American Joint Committee on Cancer tumor–node–metastasis (TNM-UICC/AJCC) classification (8th edition). The surgical treatment was a superficial parotidectomy or total parotidectomy for any tumor located in the parotid gland; in particular, a superficial parotidectomy was performed only for T1 or T2 low-grade superficial tumors, while a total parotidectomy was performed for high-grade or T3-T4 tumors [13]. Submandibular sialoadenectomy was performed for any tumor located in the submandibular gland and excision with a healthy margin of 1 cm was performed for any tumor located in the minor salivary glands. On the basis of the clinical, radiological, and cytological/histopathological analysis, neck dissection was performed in selected cases in accordance with the literature [14]. In particular, for cN0 T1 or T2 low-grade cancer, a wait-and-see approach was performed, whereas in cN0 patients with high-grade cancers or low-grade T3 or T4 tumors, elective neck dissection was performed. In cN+ patients, a Modified Radical Neck Dissection was performed. [13,14] According to the literature [13], post-operative radiotherapy was performed in the following cases: High-grade tumors, positive margins; perineural invasion; lymph node metastases; lymphatic or vascular invasion; and T3-T4 tumors. In particular, all the patients in the advanced AJCC stages (III/IV) received post-operative radiotherapy; patients in early AJCC stages (I/II) received post-operative radiotherapy only in the aforementioned conditions. 

Hematological parameters such as neutrophil count, platelet count, monocyte count, and lymphocyte count were measured within one week before the surgery. The NLR was defined as the neutrophil count divided by the lymphocyte count; the PLR was obtained by dividing the platelet count by the lymphocyte count; the SII was defined as the neutrophil count multiplied by the platelet count divided by the lymphocyte count; and the SIRI was defined as the neutrophil count multiplied by the monocyte count divided by the lymphocyte count.

### 2.3. Follow-Up Investigation

We followed up by means of a post-operative review registration and telephone investigation. Post-operative follow-up evaluations were performed every six months up to two years and then thereafter every year or until death. The latest follow-up evaluation was conducted in February 2022. The routine follow-up assessment included a physical examination, laboratory testing, neck ultrasonography, and/or a maxillofacial/neck dynamic computed tomography (CT) scan. OS was calculated from surgery to death from any cause or to the date of the latest follow-up.

### 2.4. Statistical Analysis

Receiver operating curves (ROCs) were constructed and the corresponding areas under the curves (AUCs) were computed to evaluate the performances of the PLR, NLR, SIRI, and SII markers in terms of predicting OS for the complete sample. For each marker, optimal cut-offs were determined by maximizing the Youden index and the corresponding accuracy, sensitivity, and specificity, and positive predictive and negative predictive values were provided. Additionally, computed optimal thresholds were adopted to evaluate the predictive performances of marker combinations using the ROC. For each value above the threshold, a score equal to 1 was assigned. In the case of combined indicators, a score of 2 was assigned if both single biomarkers were above the threshold, a score of 1 when only one was above the threshold, and a score of 0 when both were below the threshold. The Kaplan–Meier method was used to estimate the survival curves of the different groups, and the differences were assessed with the log-rank test. Associations between the considered factors and the time-to-event outcome variable (OS) were evaluated using univariate Cox regression models. Factors that proved to be significant in the simple regression analysis were additionally added to a multiple logistic regression model. For all the analyses, the significance level was set at α = 0.05. All the analyses were performed using the statistical software R, version 4.0.3. 

## 3. Results

### 3.1. Patient Characteristics

A total of 74 patients were included in this study. The main features of the patients are shown in Table 1. Of these patients, 38 (51.4%) were male and 36 (48.6%) were female, with a median age of 56 years (interquartile range (IQR): 13–85 years). The tumors were located in the parotid gland in 49 patients (66.2%), in the minor salivary glands in 15 patients (20.3%), and in the submandibular gland in 10 patients (13.5%). The size of the tumor was more than 4 cm in 11 patients (15%); nevertheless, none of these patients showed involvement of the facial nerve; so cT1 was observed in 22 patients (30%), cT2 was observed in 41 (55%), while cT3 was observed in 11 (15%). Fifty-three patients (72%) showed a cN0; a neck dissection was performed in 24 cases (32.4%), with 21 of these resulting positive for lymph node metastasis; so 21 patients were found pN+ while 3 patients were found pN0. Moreover, a total of 20 patients were at AJCC TNM stage I (27%), 28 at stage II (38%), 23 at stage III (31%), and 3 at stage IV (3%). The main histopathological types are reported in Table 2, with the most frequent types being adenocarcinoma (17.6%), mucoepidermoid cancer (17.6%), and adenoid cystic carcinoma (16%). In particular, the most frequent type in the parotid gland was mucoepidermoid carcinoma (18.3%), and in the submandibular gland, it was adenoid cystic carcinoma (33%). In the minor salivary glands, there was the same number of cases of these three tumors (26.6%).

Post-operative radiotherapy was performed in 33 patients (44,6%): 26 patients were in advanced AJCC stages (III/IV) and 7/48 patients were in early stages (I/II). 

### 3.2. Optimal Cut-Off Values for the Biomarkers (NLR, PLR, SII and SIRI)

A ROC analysis was performed to identify the optimal threshold with the highest sensitivity and specificity in predicting mortality. The resulting values were 3.95 for the NLR, 187.6 for the PLR, 917.585 for the SII, and 2.045 for the SIRI (sensitivity and specificity, respectively: 0.78 and 0.89 for the NLR, 0.56 and 0.91 for the PLR, 0.83 and 0.93 for the SII, and 0.78 and 0.88 for the SIRI, respectively). The raw accuracy was assessed to be 0.86 for the NLR, 0.82 for the PLR 0.91 for the SII, and 0.85 for the SIRI. Accordingly, we decided to exclude the PLR in this study as it was the marker with the lowest predictive performance (Figure 1A,B).

The median follow-up time was 74.5 months (IQR: 6–135 months). During the follow-up period, the death prevalence was 18 (24.0%) patients, meaning that 56 (76.0%) pf patients survived at the latest follow-up; of these 18 patients, 15 belonged to the advanced stage (III-IV) and 3 patients to the early stage (I-II). Three patients died of cardiovascular events, three patients died of unknown diseases, and twelve patients died of salivary gland tumors.

The overall survival at 1, 3, and 5 years estimated on time-to-event data was, respectively, 91.9%, 85.1%, and 77%. Seventeen patients died in the first five years of follow-up. The simple Cox regression analysis revealed, excluding gender, tumor location, smoking, and BMI, that the NLR, PLR, SII, and SIRI were statistically significant (*p* < 0.05) as were age, AJCC TNM stage, tumor size, lymph node metastasis, and adjuvant radiotherapy (Table 3). In the multiple Cox regression analysis, we found that the NLR (*p* = 0.661, HR = 0.57, 95% CI 0.05–6.91), PLR (*p* = 0.744, HR = 1.20, 95% CI 0.41–3.51), and SIRI (*p* = 0.164, HR = 4.19, 95% CI 0.56–31.6) were not significant. In contrast, the SII (*p* = 0.007), tumor size (*p* = 0.021), and lymph node metastasis (*p* = 0.038) were statistically significant (Figure 2A–C).

Based on the results of the above ROC analysis, we combined the inflammatory indicators with the best accuracy (the NLR, SIRI, and SII) to establish three prognosis score factors, which were designated as NLR + SII, NLR + SIRI, and SII + SIRI.

The score range of the three score factors was from 0 to 2, with 0 defined as the absence of any marker over the optimal threshold, 1 defined as one of the two factors above the optimal threshold, and 2 defined as both factors over the optimal threshold. The estimated 1-year, 3-year, and 5-year OS probability of the NLR+ SII combination with a score of 2 was 64.7%, 41.2%, and 17.7%, respectively. The estimated 1-year, 3-year, and 5-year OS probability of the NLR+ SIRI combination with a score of 2 was 66.7%, 44.4%, and 22.2%, respectively. The estimated 1-year, 3-year, and 5-year OS probability of the SII + SIRI combination with a score of 2 was 62.5%, 37.5%, and 12.5%, respectively (Figure 2A–C).

As shown in Figure 3A,B, the Kaplan–Meier curve indicated that there were significant differences in the 5-year OS of MSGT patients at different levels of the three combined prognostic indicators. High scores were associated with a poorer prognosis. The estimated 5-year OS probability in patients with SII + SIRI scores of 0, 1, and 2 was 96%, 87.5%, and 12.5%, respectively (*p* < 0.001). The estimated 5-year OS probability in patients with NLR + SIRI scores of 0, 1, and 2 was 94.1%, 100%, and 22.2%, respectively (*p* < 0.001). Finally, the estimated 5-year OS probability in patients with NLR + SII scores of 0, 1, and 2 was 96.2%, 80%, and 17.7%, respectively (*p* < 0.001) (Figure 2A–C). Based on the results of the ROC curves, the combination that achieved the highest predictive value, with an AUC = 0.884, was SII + SIRI (Figure 3A,B).

## 4. Discussion

MSGTs are rather rare neoplasms, accounting for 1–5% of all head and neck cancers. The clinical management of MSGTs is still a challenge for clinicians, with the direct and indirect costs for the diagnosis and treatment of these tumors being significant. Jacobson et al. have shown that the overall cost burden for salivary gland cancers in the USA amounts to approximately $96,520–$153,892 for each patient according to the treatment complexity [13,15]. For this reason, the need to develop a fast and low-cost prognostic system to stratify patients at a higher risk appears to be mandatory.

Several authors have already demonstrated the efficacy of inflammatory biomarkers as easily available and low-cost prognostic values in various fields of oncology. In particular, in relation to head and neck cancers and in oral cancer, for example, high levels of inflammatory blood biomarkers such as the NLR and PLR are potential markers of poor OS and Disease-Free Survival (DFS) [10,16]. In fact, considering that the inflammatory microenvironment has also been proposed as an influencing factor in cancer, the NLR and PLR can reflect the relationship between the inflammatory activating factor and regulatory factor. Increased neutrophil counts and/or decreased lymphocyte counts lead to an increased NLR. Neutrophils are often distributed in the tissues surrounding tumors, where they secrete large quantities of vascular endothelial growth factor, providing an appropriate microenvironment for the promotion of local tumor invasion and metastasis [17].

Lymphocytes, instead, are major anticancer factors. Lymphocytes play a specific role in killing tumor cells by inducing cytotoxic cell death and cytokine production to mediate host immune responses, thereby inhibiting tumor cell proliferation. Nevertheless, in a proinflammatory tumor state, neutrophils can suppress the effective immune response mediated by lymphocytes [18].

Moreover, tumor cells can activate platelets and stimulate aggregation, which can protect the neoplasia from the attack of lymphocytes. Therefore, platelets may also have a significant impact on tumorigenesis and metastasis [19,20]. Since the NLR and PRL are composed of only two inflammatory cells, the prediction ability of these biomarkers for survival outcomes may not be the best. For this reason, additional inflammatory indices have recently been developed. The SII is based on three parameters, neutrophils, platelets, and lymphocytes, while the SIRI consists of three types of inflammatory cells, neutrophils, monocytes, and lymphocytes [9]. Furthermore, numerous studies have shown that the predictive ability of the SIRI is more powerful than that of other inflammatory biomarkers, such as the NLR, PLR, and mixed lymphocyte reaction (MLR) [21,22]. Based on these findings, many authors have investigated how the inflammatory status may be related to the prognosis of MSGT patients. Fridman et al. have shown that the CD8+ higher tumor-infiltrating lymphocyte (TIL) density in the tumor microenvironment is a favorable prognostic indicator in patients with salivary gland cancer [23]. Sato et al. have conducted a univariate and multivariate analysis on the role of the tumor proportion score (TPS) in MSGT patients: The authors have revealed that the univariate analysis showed that the T and N classification, histological grade, lymphatic invasion, venous invasion, perineural invasion, and TPS were statistically significant predictive factors of OS and DFS, while the multivariate analysis showed that the TPS was statistically more significant when associated with a high tumor grade, positive lymphatic invasion, and positive perineural invasion [24]. The increase in the predictive capacity of these indices, when combined, has already been demonstrated in other fields of oncology, for example, in relation to gastric cancer, lung cancer, and oral cancer [16,25,26]. To the best of our knowledge, there is no evidence in the literature regarding any correlation between a high level of inflammatory biomarkers (the NLR, PLR, SII, and SIRI), alone or combined, and a poorer prognosis in MSGT patients. In our study, the univariate analysis showed that all the markers studied (the NLR, PLR, SII, and SIRI), when above the threshold, were significant in identifying a worse prognosis. Considering the results of the ROC analysis, of the inflammatory biomarkers, the PLR resulted in a lower accuracy (0.82) and therefore was excluded from the analysis. To provide a practical application of these findings, we combined the inflammatory indicators with the best accuracy (the NLR, SIRI, and SII) to establish three prognosis score factors, which were designated as NLR + SII, NLR + SIRI, and SII + SIRI. In our sample, based on the results of the ROC curves, the best combination, with an AUC = 0.884, was SII + SIRI. This combination has the capacity to fully assess the balance between the host immune and inflammatory conditions by consolidating all three cells simultaneously in the formula. Nowadays, the severity of the disease on the TNM scale, and in particular, the T parameter, is the most important independent factor that worsens the prognosis in MSGT patients [27,28]. Nevertheless, according to our results, patients with the same disease stage seem to have a different prognosis according to the SII + SIRI score. It is noteworthy that among the patients in stage 3 (23 patients), 100% of those with a score equal to 2 (10 patients) died in the first five years of follow-up. Of the 13 patients with a score of either 0 or 1, only 2 (15%) died, one with a score of 1 dying at 38 months and the other with a score equal to 0 dying after five years of follow-up. For these reasons, it is reasonable to combine the SII and SIRI to obtain a scoring system useful for stratifying patients at a higher risk. This information may provide a scientific basis for clinical treatment and follow-up strategy adjustment, in the sense that patients with a poor prognosis in the same clinical stage can be easily evaluated in terms of their SII + SIRI score, an approach that is equivalent to other auxiliary treatments and close follow-up. This information can have clinical implications in terms of treatment decision making: In particular, in the early stages, the evaluation of carrying out neck dissection or radiotherapy, while in the advanced stages, the evaluation of adding a targeted therapy or anti-inflammatory therapies to the already known therapies. This analysis could be the object of future prospective studies. There are some potential limitations in relation to this study. First, this was a retrospective analysis that included 74 MSGT patients, and all the data are from one medical institution. Secondly, unknown inflammatory pathologies missed during the anamnesis could have generated a selection bias. Furthermore, adjuvant RT and CHT treatment represents a selection bias in our sample. In AJCC stages I and II, only some patients underwent radiotherapy. This creates inhomogeneity in the sample while it certainly has no effect in stages III and IV since all patients underwent treatment.

Nevertheless, these preliminary results from this current retrospective study can be used as a theoretical basis for the next multicenter study on a large-scale population.

## 5. Conclusions

In conclusion, the encouraging results of our study show that the combination of the SII and SIRI scores can be used as a supplement to the TNM stage in pre-operative risk stratification to effectively guide the treatment strategy and post-operative follow-up of MSGT patients. Both the SII and SIRI are easily available indicators in routine pre-operative examinations, making measures economical, reproducible, and repeatable.

## Figures and Tables

**Figure 1 cancers-14-05934-f001:**
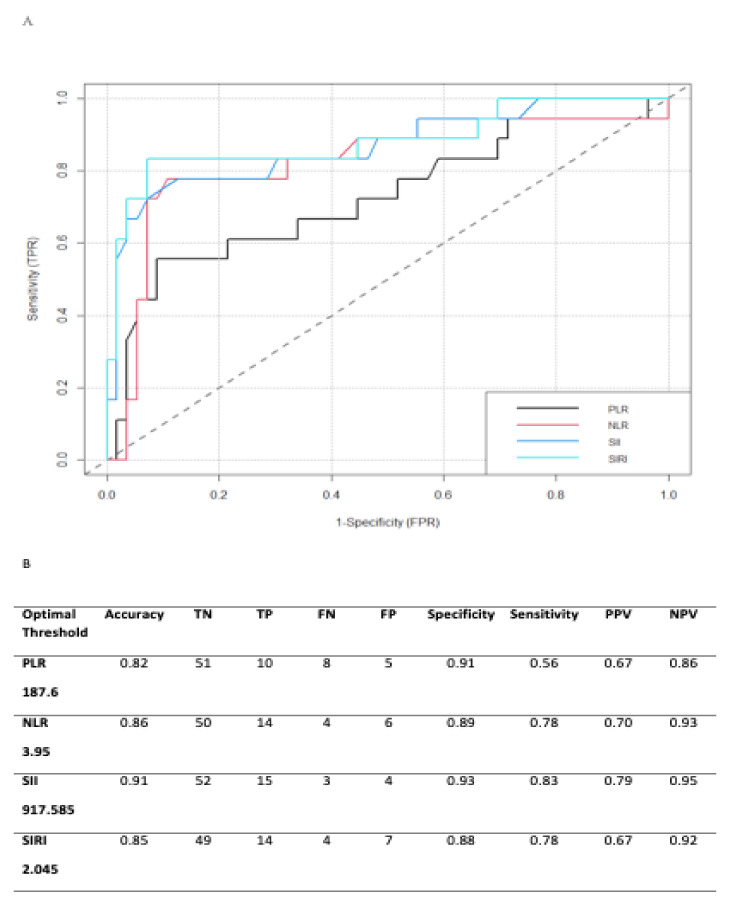
(**A**) ROC Curves of all markers, (**B**) best thresholds of all markers.

**Figure 2 cancers-14-05934-f002:**
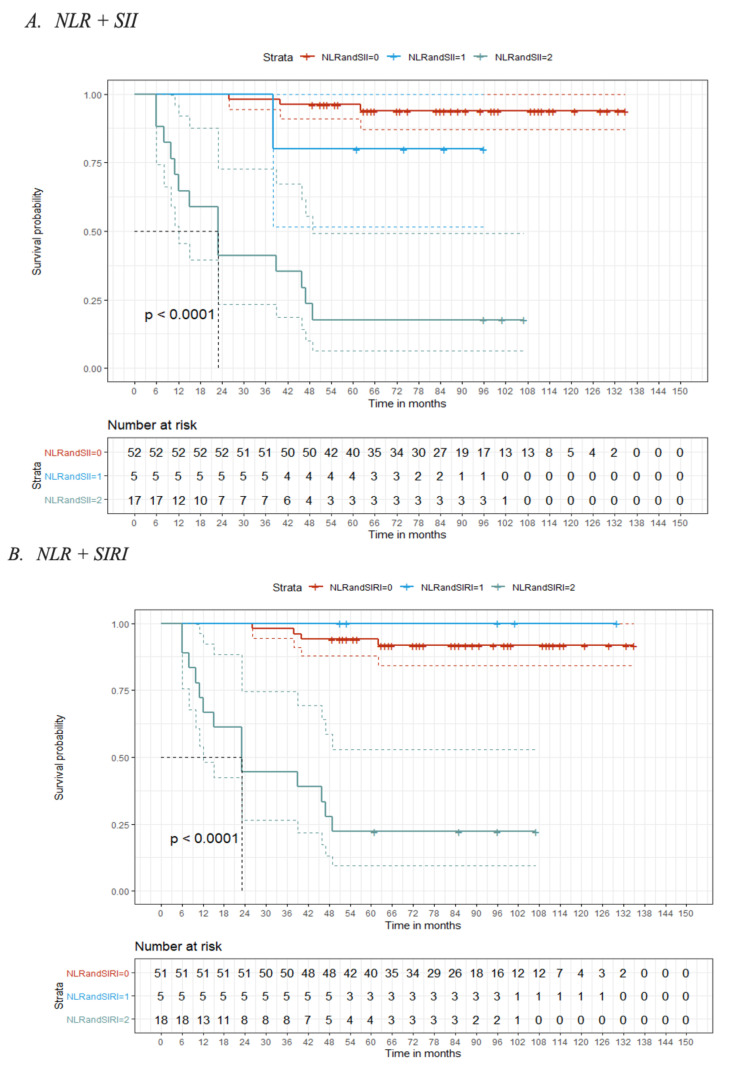
Kaplan–Meier curves. (**A**) NLR + SII; (**B**) NLR + SIRI; (**C**) SII + SIRI.

**Figure 3 cancers-14-05934-f003:**
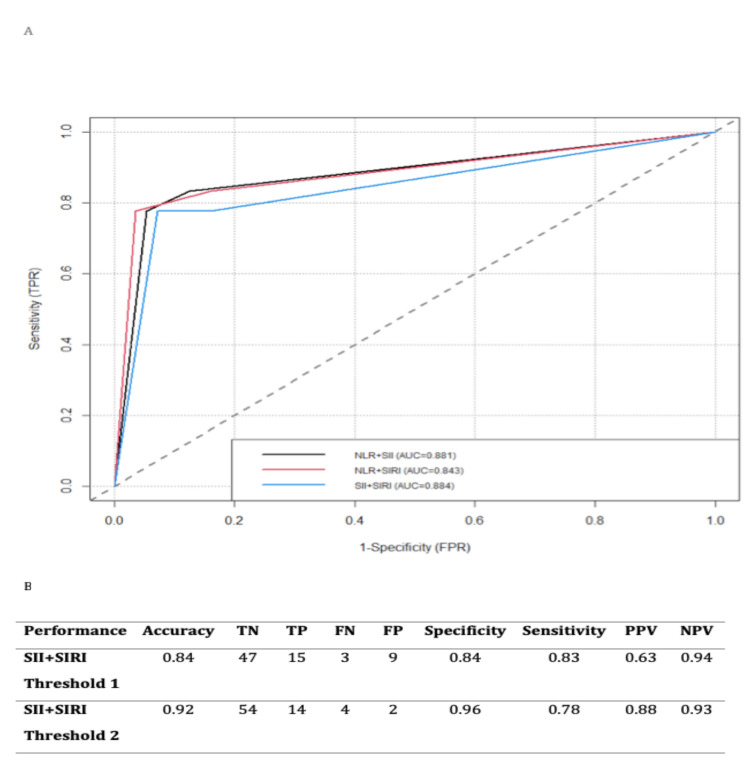
(**A**) ROC Curves with all marker combinations; (**B**) best threshold of SII + SIRI.

**Table 1 cancers-14-05934-t001:** General population features.

Variables	Total Cases 74
Age (years)	
≤60	45 (61%)
>60	29 (39%)
Gender	
Female	36 (48%)
Male	38 (52%)
Tumor location	
Major salivary glands	59 (80%)
Minor salivary glands	15 (20%)
cTstatus	
T1	22 (30%)
T2	41 (55%)
T3	11 (15%)
cNstatus	
cN−	53 (72%)
cN+	21 (28%)
pTstatus	
T1	22 (30%)
T2	41 (55%)
T3	11 (15%)
pNstatus (on 24 cases)	
N0	3 (12.5%)
N1	18 (75%)
N2a	2 (8.5%)
N2b	1 (4%)
AJCC TNMstage	
I	20 (27%)
II	28 (38%)
III	23 (31%)
IV	3 (3%)

**Table 2 cancers-14-05934-t002:** Main histopathological types.

Tumor Localization	N° Cases	Histopathological Types
Parotid Gland	49	Mucoepidermoid Cancer 9 (18.4%)
		Adenocarcinoma 8 (16.3%)
		Squamous Cell Carcinoma 6 (12.2%)
		Adenoid Cystic Carcinoma 5 (10.2%)
		Myoepithelioma 5 (10.5%)
		Carcinoma ex pleomorphic adenoma 5 (10.5%)
		Mammary analog secretory carcinoma 3 (6.1%)
		Acinar Adenocarcinoma 2 (4.1%)
		Microcystic adnexal carcinoma 1 (2%)
		Intraductal Carcinoma 1 (2%)
		Undifferentiated carcinoma 1 (2%)
		Sarcoma 1 (2%)
		Lymphoepithelial Carcinoma 1 (2%)
		B-cell lymphoma 1 (2%)
Submandibular Gland	10	Adenoid Cystic Carcinoma 3 (30%)
		Carcinoma ex pleomorphic adenoma 2 (20%)
		Squamous Cell Carcinoma 2 (20%)
		Carcinosarcoma 1 (10%)
		Sarcoma 1 (10%)
		Adenocarcinoma 1 (10%)
Minor salivary Glands	15	Adenocarcinoma 4 (27%)
		Adenoid Cystic Carcinoma 4 (27%)
		Mucoepidermoid Cancer 4 (27%)
		Carcinoma ex pleomorphic adenoma 1 (6.7%)
		Myoepithelioma 1 (6.7%)
		Clear-cell carcinoma 1 (6.7%)
TOTAL	74	Adenocarcinoma 13 (17.6%)
		Mucoepidermoid Cancer 13 (17.6%)
		Adenoid Cystic Carcinoma 12 (16%)
		Carcinoma ex pleomorphic adenoma 8 (10.8%)
		Squamous Cell Carcinoma 8 (10.8%)
		Myoepithelioma 6 (8%)
		Mammary analog secretory carcinoma 3 (4%)
		Acinar Adenocarcinoma 2 (2.7%)
		Sarcoma 2 (2.7%)
		Microcystic adnexal carcinoma 1 (1.4%)
		Intraductal Carcinoma 1 (1.4%)
		Undifferentiated carcinoma 1 (1.4%)
		Lymphoepithelial Carcinoma 1 (1.4%)
		B-cell lymphoma 1 (1.4%)
		Clear-cell carcinoma 1 (1.4%)
		Carcinosarcoma 1 (1.4%)

**Table 3 cancers-14-05934-t003:** Cox regression of all markers with the best thresholds.

	Univariate				Multivariate		
Characteristic	N	HR	95% CI	*p*-Value	aHR	95% CI	*p*-Value
Sex	74			0.053			
F		—	—				
M		2.62	0.93, 7.34				
Age	74			0.029			0.917
≤60		—	—		—	—	
>60		2.83	1.09, 7.31		0.94	0.30, 2.92	
AJCC TNM_Stage	74			<0.001			
I		—	—		—	—	
II		0.86	0.12, 6.12		0.78	0.10, 5.97	0.808
III		7.81	1.75, 35.0		2.35	0.27, 20.8	0.442
IV		15.4	2.16, 110		3.35	0.26, 43.5	0.355
TUMOR_LOCATION	74			0.290			
major		—	—				
minor		0.48	0.11, 2.11				
TUMOR_SIZE	74			0.001			0.021
≤4		—	—		—	—	
>4		5.65	2.18, 14.6		3.53	1.21, 10.3	
Lymphnode_status	74			<0.001			0.038
N-	53	—	—		—	—	
N+	21	8.68	3.08, 24.5		3.85	1.08, 13.8	
ADJUVANT_RADIOTHERAPY	74			<0.001			0.159
0	41	—	—		—	—	
1	33	5.58	1.83, 17.0		2.85	0.66, 12.2	
Smoke	74			0.671			
0	54	—	—				
1	20	0.79	0.26, 2.40				
BMI	74	1.01	0.90, 1.13	0.843			
NLRevent	74			<0.001			0.661
≤3.95		_	_		_	_	
>3.95		16.2	5.29, 49.7		0.57	0.05, 6.91	
PLRevent	74			<0.001			0.744
≤187.6		—	—		—	—	
>187.6		7.92	3.10, 20.3		1.20	0.41, 3.51	
SIIevent	74			<0.001			0.007
≤917.585		—	—		—	—	
>917.585		28.0	7.98, 98.0		15.7	2.11, 117	
SIRIevent	74			<0.001			0.164
≤2.045		—	—		—	—	
>2.045		15.0	4.87, 46.2		4.19	0.56, 31.6	

## Data Availability

Not applicable here.

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
