# Peer review of "The Combination of Inflammatory Biomarkers as Prognostic Indicator in Salivary Gland Malignancy"

_cancers, 2022, doi:10.3390/cancers14235934_

Round 1

Reviewer 1 Report

The current manuscript addresses the lack of effective pre-surgery prognostic biomarkers for malignant salivary gland tumors (MSGT). A retrospective analysis was performed to investigate if SII and SIRI along with NLR and PLR can be used individually or in combination to predict the overall survival of patients who underwent surgery for MGST. The results are presented well with right controls and data was segregated at best for a single site study in a specific date range. The study describes the potential usage of these biomarkers for better patient care. However, also identify the limitations of this study and warrant further investigation with larger cohorts and a multiple-site study. I recommend that the current manuscript can be accepted with few suggested yet important revisions.

Minor changes:

1.     Font consistency is required for all figures. Figure 3 needs to be altered to increase the font size

2.     Spacing check, grammar check is required throughout the manuscript.

Author Response

Reviewer n° 1: Dear Sir and Colleagues, thank you very much for your suggestions and queries regarding our paper. It can be remarkably improved.

  1. Thank you for the suggestion, Font consistency was added in all figures and we divided the figure 2 in 3 different imagines to improve the resolution.

  1. Thank you for the suggestion, spacing check and grammar check have been revised.

We hope that the revised version can be enough and more interesting for you according to the changes you have requested.

Reviewer 2 Report

Dear authors,

thank you for your work and the interesting article. I agree that salivary gland malignancies are difficult to treat, and prognostic parameters would be of great help. The more I appreciate your efforts to elucidate the impact of serum parameters in the field.

Unfortunately, the present article has several weaknesses, although it is clear that solid data is hard to obtain due to the heterogeniety of these tumors are their scarcity. I may share with you a few points I stumpled upon while reading your manuscript.

Title: In my opinion, the title should read: „The combination of inflammatory biomarkers as prognostic indicator in salivary glands malignancy“

Line 28: „malignant salivary glands tumors“ should read „malignant salivary gland tumors“.

Line 43: „…can influence“ should be replaced with „are associated with“, as there is no proof that these changes are the real cause of altered OS rates.

Line 76: „pathological stage“ should read for example „UICC stage“ or similar.

Line 99: should read „malignant salivary gland tumor“.

Line 116f: For malignant tumors, superficial parotidectomy is a therapy at risk and only acceptable for small, certain low grade tumors. In case that superficial parotidectomy was conducted, this group may include smaller, low graded tumors which are than compared with tumors of larger size or higher gradings, or with a more aggressive histology. This may create a bias.
Further, this may also apply for indications for neck dissections – no indication is given when and why and in which tumor entities a neck dissection was performed.

Line 166f: the authors state that neck dissections were performed in 24 cases, so it must be assumed that 50 patients presented with a cN0 neck status and did not receive a neck dissection. The patients who did not receive a neck dissection cannot be assigned a pN-status (as it is done in Table 1). In detail, 21 of 24 neck dissection specimen were found with positive lymph nodes. In Table 1, it is given that the category „lymph node metastasis“ mixes clinical negative lymph nodes (cN0) with proven ones (pN0). This is not precise. It seems that 24 of 74 patienst received a neck dissection. 21 patients were found as pN+ (is is not specified to which extent in the text, except in Table 1), 3 were found pN0, and 50 patients were considered as cN0 (although not specified – then, the information in Table 1 is incorrect, as all patients were assigned a pNX status, but not all received a neck dissection). Please correct and specify this.

Line 165: „The size of the tumor was more than 4cm in only 11 patients…“. This is not precise. Please provide a T-category (as in Table 1), as size is not the only criterion (extracapsular growth, facial nerve involvement, etc.). It is surprising that no tumor seemed to have affected the facial nerve in 49 cases oft he parotid gland, as this would be a T4-category.

Line 161f: please use the term „mucoepidermoid carcinoma“ instead of „cancer“.

Line 168: TNM provides categories, not stages. UICC provides stages. Please adapt.

Line 173/174: What does the sentence in lines 173/174 mean? Please specify. How many patients received adjuvant radio-(chemo)-therapy? Were they mixed in the same group as those without adjuvant treatment?

Line 189: Why did these patients die? Due to the tumor, due to other reasons? Please specify.

Line 203f: What do the authors want to express in this paragraph? It seems that in the first part, they give the estimated OS probability they calculated by using the ROC curves and markers, or a combination of these. In the second part, the OS survival rates received by Kaplan-Meier curves are analyzed. Why are these values estimated, as they should represent real death cases?
When calculated probabilities are compared with the real Kaplan-Meier data, it is obvious that these differ quite much from another. Does this mean that the estimated values are different from the real ones? How precise ist he estimation due to the analyzed markers? Please explain.

Line 305-307: Where do the authors show that patients with the same T-category have a different prognosis in OS? Did the authors compare the same tumor entities?

Line 313: What are the clinical consequences in patients with a lower prognosis? Elective neck dissection, adjuvant therapy?

Line 317f: From my point of view, the largest limitation of the present study is that many different tumor entities have been mixed up in one cohort. This also accounts for the different T-categories and adjuvant therapy, which has not been given in detail. Further, clinical and pathological results are mixed up (cN-status, pN-status). This creates a cohort that is very inhomogenous. Given that, conclusions that can be drawn here are very limited.   

In summary, the article is interesting and structured. However and first, several passages in the text are confusing and should be properly revised, important information is missing. Second, the article mixes up pathohistologic proven and clinically diagnosed patients with one another, does not recognize adjuvant therapies, and does not consider different tumor entities and their behavior.
It is understandable that, respecting that there are many different tumor entities in salivary glands which behave very differently, a large cohort of patients is necessary to figure out details. However, the article at least seeks to do so, and this is not sufficient in the present form. Therefore, it is very important that used data are properly assigned, which seems not to be true in this present article.

Author Response

Reviewer n° 2: Dear Sir and Colleagues, thank you very much for your suggestions and queries regarding our paper. It can be remarkably improved.

1.Thank you for the observation, we changed the title in The combination of inflammatory biomarkers as prognostic indicator in salivary glands malignancy

  1. Thank you for the observation, we changed in line 28: „malignant salivary glands tumors“ to malignant salivary gland tumors“.

  1. Thank you for the suggestion, we changed in line 43: „…can influence“ to “are associated with”

  1. Thank you for the suggestion in line 76: „pathological stage“ has been changed in “AJCC stage“ according toAmerican Joint Committee on Cancer stage , furthermore we mentioned the new reference in the references section

  1. Thank you for the suggestion, we changed in line 99: „malignant salivary gland tumor“.

  1. Thank you for the suggestion, in line 126f according to ASCO Guideline, we performed superficial parotidectomy only for T1 or T2 low grade superficial tumors, while total parotidectomy was performed for high-grade or T3-T4 tumors.Furthermore for cN0 T1 or T2 low-grade cancer a wait-and-see approach was performed, whereas in cN0 patients with high-grade cancers or low-grade T3 or T4 tumors an elective neck dissection was performed. In cN+ patients, a Modified Radical Neck Dissection was performed. We mentioned the new reference in the references section.

  1. Thank you for the suggestion, we revised the table 1 and we specified that the patients in cN0 were 53 (72%) and the pNstatus was calculated only on 24 patients that performed the neck dissection. A neck dissection was performed in 24 cases (32.4%), with 21 of these resulting positive for lymph node metastasis; so 21 patients were found pN+ while 3 patients were found pN0

  1. Thank you for the suggestion, we revised the table 1: The size of the tumor was more than 4 cm in 11 patients (15%),nevertheless, none of these patients showed involvement of the facial nerve; so cT1 were 22 patients (30%), cT2 were 41 (55%), while cT3 were 11 ( 15%).

53 patients (72%) showed a cN0; a neck dissection was performed in 24 cases (32.4%), with 21 of these resulting positive for lymph node metastasis; so 21 patients were found pN+ while 3 patients were found pN0. Moreover, a total of 20 patients were at AJCC TNM stage I (27%), 28 at stage II (38%), 23 at stage III (31%) and 3 at stage IV (3%).

  1. Thank you for the suggestion in line 161f: we used the term „mucoepidermoid carcinoma“ instead of „cancer“.

  1. Thank you for the suggestion in line 168 we used the AJCC TNM

  1. Thank you for the suggestion in line 173/174, thank you for pointing this out, for sure adjuvant RT and CHT treatment represents a selection bias in AJCC stages I and II, while it certainly has no effect in stages III and IV since all patients underwent treatment. Given the critical circumstances, we proceeded to include this aspect within the limits of the study. We affirmed according with literature when post-operative radiotherapy was performed: high-grade tumors, positive margins, perineural invasion, lymph node metastases, lymphatic or vascular invasion and T3-T4 tumors. In particular, all the patients in the advanced AJCC stages (III/IV) received post-operative radiotherapy; patients in early AJCC stages (I/II) received post-operative radiotherapy only in the aforementioned conditions. Post-operative radiotherapy was performed in 33 patients (44,6%): all the 26 patients in advances AJCC stages (III/IV) and 7/48 patients in early stages (I/II).

  1. Thank you for the suggestion in line 189 we mentioned that during the follow-up period, 18 (24.0%) patients died, meaning that 56 (76.0%) were alive at the latest follow-up; of these 18 patients 15 belonged to the advanced stage (III-IV) and 3 patients to the early stage ( I-II). 3 patients died of cardiovascular event, 3 patients died of unknown diseases and 12 patients died of salivary gland tumours.

13 We thank the reviewer for the comment. We have noticed that the paragraph was not clear in differentiating which OS estimates were produced on the prevalence rates and which were instead obtained on time-to-event data and thus with the Kaplan-Meier method. We have rephrased and clarified the paragraph specifying which OS rate was estimated as prevalence (i.e. as a rate, a percentage) and which OS data were instead estimated on time-to-event data, using the Kaplan-Meier method.

14 Thank you for the suggestion in line 305 , the risk of the T-category on the OS can be evidenced from the Univariate Cox regression model (table 3), where it is shown how the estimated risk changes by T-stage (this information is reported as Hazard ratio with the 95% confidence interval). The scope of the paper is to analyze the prognostic performance of the inflammatory markers, thus to adjust for the T-category factor we have included this predictor in both the univariate and multivariate Cox regression model.

15 Thank you for the suggestion in line 313 we clarify which could be the clinical consequences in patients with a lower prognosis. This information can have a clinical implication in terms of treatment decision making: in particular in the early stages the evaluation of carrying out neck dissection or adjuvant radiotherapy, while in the advanced stages the evaluation of adding a target therapy or anti-inflammatory therapies to the already known therapies. This analysis could be the object of future prospective studies.

16  We thank the reviewer for the points raised. We have made several corrections through the whole manuscript in order to address many of the points raised in this review. First of all, in order to prove that the prognostic performance of the inflammatory markers was comparable across the T-categories of the tumors, we have added the ROC and performance measures of all the markers stratified by T-stage (in particular, we have created a group for T-stages I and II and a group for T-stages III and IV)[ see supplementary material]. As can be evidenced by the results, there is no major difference in the performances of the markers across these two groups. Concerning the impact of the T-stage factor on the OS rate, this factor was accounted for in the univariate and multivariate Cox regression models, where the HRs were adjusted for all the factors included, as shown in Table 3. Thus, all the risks are estimated taking into account the T-category as well, and how the risk changes by stage can be read from the HR and their confidence intervals from the Univariate table. Considering the different adjuvant therapy, we have added the fact that this information was not included as a limitation of the present study

We hope that the revised version can be enough and more interesting for you according to the changes you have requested.
